# Recent Progress and Challenges for Drug-Resistant Tuberculosis Treatment

**DOI:** 10.3390/pharmaceutics13050592

**Published:** 2021-04-21

**Authors:** Filia Stephanie, Mutiara Saragih, Usman Sumo Friend Tambunan

**Affiliations:** Bioinformatics Research Group, Department of Chemistry, Faculty of Mathematics and Natural Sciences, Universitas Indonesia, Depok 16424, Indonesia; filia.stephanie@sci.ui.ac.id (F.S.); mutiara.saragih@sci.ui.ac.id (M.S.)

**Keywords:** tuberculosis, drug-resistance, drug discovery and development, host-directed therapy

## Abstract

Control of *Mycobacterium tuberculosis* infection continues to be an issue, particularly in countries with a high tuberculosis (TB) burden in the tropical and sub-tropical regions. The effort to reduce the catastrophic cost of TB with the WHO’s End TB Strategy in 2035 is still obstructed by the emergence of drug-resistant TB (DR-TB) cases as result of various mutations of the MTB strain. In the approach to combat DR-TB, several potential antitubercular agents were discovered as inhibitors for various existing and novel targets. Host-directed therapy and immunotherapy also gained attention as the drug-susceptibility level of the pathogen can be reduced due to the pathogen’s evolutionary dynamics. This review is focused on the current progress and challenges in DR-TB treatment. We briefly summarized antitubercular compounds that are under development and trials for both DR-TB drug candidates and host-directed therapy. We also highlighted several problems in DR-TB diagnosis, the treatment regimen, and drug discovery that have an impact on treatment adherence and treatment failure.

## 1. Introduction

*Mycobacterium tuberculosis* (MTB) is the etiological agent of tuberculosis (TB), one of the leading causes of death due to an infectious agent according to the WHO. This disease results in 1.5 million fatalities annually. TB cases are centralized in 30 high TB burden countries, which account for 87% of all cases. India, China, Indonesia, the Philippines, Pakistan, Nigeria, Bangladesh, and South Africa lead the count for this health crisis [1]. MTB infection can manifest as two clinical conditions, i.e., pulmonary TB and extrapulmonary TB. The former is most common (80% of total cases in estimation) and can result in death through chronic respiratory failure [2]. In several cases, stimulation of immune response during MTB infection creates a repressed environment for MTB growth and results in no clinical manifestation. This condition called latent TB, where the bacteria become inactive but are still alive. Latent-TB patients acquire a higher risk for TB development in the future [3]. 

MTB is classified in the *Mycobacterium* complex group, along with *M. bovis, M. microti, M. canettii, M. africanum*, etc. [4]. Sequence analysis of MTB H37Rv and annotation of the coding/non-coding region has been done, resulting in the characterization of several genes and proteins responsible for the MTB life cycle [5]. Based on the genomic study, there are more than 200 genes (6%) in the MTB genome that are needed to code proteins for fatty acid metabolism [6]. This indicates the importance of fatty acids in MTB physiology. One of main features of MTB is its unique cell membrane structure that can act as an impermeable barrier to foreign substances. The MTB cell membrane consists of an asymmetrical lipid bilayer made of long-chain fatty acids on the inside (mycolic acid), with a glycolipid and wax component on the outside. These inner and outer membranes create a periplasmic environment with a peptidoglycan layer that interacts covalently with mycolic acid. This cell membrane is essential for MTB biological activity, as shown by two first-line drugs for TB, isoniazid and ethambutol, which act as inhibitors of the mycolic acid synthesis pathway in MTB [7]. 

Efforts toward drug discovery and development for tuberculosis have taken place since the middle of 20th century [8]. At first, TB treatment involved antitubercular medication that is categorized according to its bactericidal efficacy. Isoniazid, pyrazinamide, ethambutol. and rifampicin were grouped into first-line drugs and were used for tuberculosis chemotherapy for years [9]. 

Nowadays, TB treatment faces a formidable obstacle due to MTB’s adaptive mechanisms toward available drugs, which result in drug resistance. MTB resistance to TB medication can be divided into several groups, such as monoresistance, polyresistance, multidrug resistance (MDR-TB), extensive drug resistance (XDR-TB), and total drug resistance (TDR-TB) [10]. Each year, among 10 million new cases of TB, 600,000 patients were identified to have MDR-TB. MDR-TB is defined as resistance to at least rifampicin and isoniazid, the most powerful antitubercular drugs [11]. Resistance to drugs in MTB can be developed through several mechanisms, such as acquired resistance, intrinsic resistance, and other special mechanisms. Acquired resistance happens in bacteria through phage-mediated horizontal gene transfer, chromosomal mutation, or antibiotics usage. While the horizontal transfer of drug resistance genes is unknown in MTB, the second mechanism, intrinsic resistance, is commonly exhibited in this pathogen and is caused by physiological reasons such as its impermeable cell wall. Other resistance mechanisms involves drug target alteration, molecular mimicry, drug structure modification, drug degradation, and drug efflux mechanisms [12].

Patients with drug-resistant TB are not susceptible to the common TB drug regimen, and this results in therapy failure. Treatment is then done using second-line agents in a complex and time-consuming regimen. A combination of injectable second-line drugs and fluoroquinolones was not effective to cure this disease [13]. Bedaquiline and delamanid are two drugs that were recently approved by U.S. FDA to broaden the treatment landscape of MDR-TB. Their bactericidal properties were examined through trials, and the newest WHO guideline for drug-resistant TB therapy listed these as an additional antitubercular drugs for short-duration drug-resistant TB treatment [14]. However, through post-marketing surveillance, more than 45% of patients showed an intolerance response toward these chemicals with severe adverse effects [15]. Until now, the efforts toward drug discovery for drug-resistant TB are still ongoing in order to find safe medications to combat this disease. This review will summarize the progress and challenges concerning the treatment of drug-resistant TB.

## 2. Progress

Traditional healers utilize local plant species to provide natural therapeutics for pulmonary TB. The most frequently listed and validated plant species recommended in TB treatment and in the management of associated diseases belong to Amaryllidaceae, Apiaceae, Asparagaceae, Asteraceae, Leguminosae, Rutaceae, and Solanaceae botanical families. Tuberculosis prevelance is correlated with malnutrion among patients, and it can impair host immune responses against MTB infection. Protein deficiency plays a role to enhance bacterial growth and dissemination due to thymic atrophy and the impaired generation and maturation of T-lymphocytes. It also impairs the protective interaction between macrophages and T-lymphocytes; higher production of transforming growth factor-β, which is a mediator of immunosuppression and immunopathogenesis in tuberculosis; decreased production of Th1 cytokines; and loss of tuberculosis resistance following BCG vaccination [16].

In this section, the current drug discoveries and development efforts for MDR-TB treatment will be discussed. MTB is considered one of the most-studied bacteria due to its pathogenicity. The whole genome of MTB H37Rv has already been sequenced for years, and knowledge of the MTB life cycle and possible drug targets have been characterized through annotation [5]. This development leads to the discovery of antitubercular compounds for consideration as TB drug candidates. In the following discussion, each drug in preclinical and clinical trials for drug-resistant TB based on their chemical classes were summarized.

### 2.1. Fluoroquinolone

Fluoroquinolone is one of the most known chemical classes in the TB second-line drug regimen. This group belongs to broad-spectrum antibiotics that have the ability to inhibit the DNA gyrase (topoisomerase II) and DNA topoisomerase IV, both of which are prominent ATP-dependent type IIA bacterial enzymes [17]. DNA topoisomerases act in several key DNA transactions, e.g., DNA gyrase plays a role as the catalyst for negative supercoiling, and topoisomerase IV facilitates chromosome segregation [18]. In the MTB genome, DNA gyrase is encoded by *gyrA* and *gyrB* for the A and B subunits of the A2B2 tetrameric protein, and topoisomerase IV coding genes *parC* and *parE* (or the) were not found. This genetic evidence makes DNA gyrase the sole target for fluoroquinolone TB drugs [19].

DNA gyrase in MTB works to catalyze the cleavage of DNA into the gate segment to facilitate the ATP-dependent transfer of another segment, while topoisomerase has a main role in regulating the topological structure of DNA [20]. Since mycobacteria only possess a single type IIA enzyme, MTB DNA gyrase carries additional responsibility to separate the chromosome, the task that originally belongs to the topoisomerase IV [21]. This dual role is different from those of other bacterial topoisomerases, and is reflected in the *gyrA* and *gyrB* identities, which only have approximately 40–44% similarity to the *Escherichia coli* gyrases [22]. This property ensures that drugs targeting DNA gyrase are more efficient in MTB than in other organisms in comparison with drugs targeting topoisomerase IV. 

Quinolone is the chemical class that has the most notable antibacterial activity with DNA gyrase as its target. These compounds inhibit the process of DNA supercoiling, leading to the stabilization of the complex through attachment of the GyrA protein to the 5′ ends of the broken DNA with covalent linkage [23]. There are several fluoroquinolone compounds that have been used for MDR-TB treatment such as moxifloxacin and gatifloxacin. These compounds are used as part of a second-line universal drug-resistant TB regimen, since they showed dissatisfactory results in the trials for first-line shortened therapy [24,25]. Moxifloxacin’s limitation presents through resistance that is developed with the occurrence of the main mutants A90V and D94G in GyrA structure. This resistance resulted in decreasing drug affinity toward the protein [26].

Levofloxacin is a fluoroquinolone tuberculosis drug that is now in a phase 2 trial for drug-resistant and latent-TB treatment. Levofloxacin was observed to have high in vitro and in vivo antituberculosis activity and is preferable compared to the other fluoroquinolones because it is more affordable [27]. This reachability makes levofloxacin easily available in high TB burden countries that have limitations in resources. The structures of moxifloxacin, gatifloxacin, and levofloxacin are visualized in Figure 1, while levofloxacin and the DNA gyrase inhibitor can be seen in Figure 2.

### 2.2. Rifamycin

Rifamycin is a group of antimicrobials that can be naturally synthesized by *Amicolatopsis rifamycinica*, but most of the medications containing rifamycin are produced artificially. This group contains the known rifamycin drugs such as rifampicin, rifabutin, rifapentine, rifalazil, and rifaximin [28]. The bactericidal activity of rifamycin toward mycobacteria is well known as several rifamycin compounds are used as therapy for many mycobacteria-related diseases such as TB and leprosy [29]. Rifampicin is one of first discovered drugs for TB treatment along with isoniazid. Rifampicin showed great affinity and was observed to be the most potent broad-spectrum antibiotic due to its ability to be diffused easily into the tissue, cells, and the pathogen bacteria [30].

MTB RNA polymerase (RNAP) is the target of rifampicin. RNAP regulates gene expression of the organism and is responsible for the transcription process in MTB cells. This protein consists of five subunits, comprising two identical a units, b, b’, and w (a2bb’w), that form the holoenzyme and is activated by the sixth subunit o (factor o), which recognizes the promoter region to initiate the transcription phase [31]. Rifampicin binds to the active site in the b subunit. The binding site in the b subunit (RpoB) is the clamp-like structure that is formed by both the b and b’ subunits. With high affinity, rifampicin interferes with the protein production in MTB, causing cell death and a curative impact in most TB cases [32]. Visualization of rifampicin in RpoB can be seen in Figure 3.

However, the usage of rifampicin as a first-line drug in TB treatment for years has caused the pathogen to gain resistance toward this compound. The resistance mechanism is promoted by the formation of mutations in the coding gene of the b subunit of RNAP, *rpoB*, mainly located in the rifampicin-resistance determining region (RRDR). The RRDR consists of 81 nucleotides in codon 507 to 533 [33]. Mutations occurring in codon 516, 526, and 531 were observed to be responsible for more than 90% of total resistant cases toward rifampicin [34]. This condition caused most of the MDR-TB and XDR-TB cases because the strains that are resistant to rifampicin also possess resistance toward isoniazid [35].

As the mutation occurred, causing an obstacle in TB therapy, the usage of rifampicin has become limited to those who are still susceptible to it. Unfortunately, more resistant cases are emerging each day, and alternatives to the dose and usage of first-line drugs should be updated. The use of rifampicin with a high dosage was tested in vivo and showed a potential to accelerate the cure for this disease [36]. This study is currently on in a phase 2 clinical trial in an effort to shorten the regimen for TB treatment. Although the mechanism is still unclear, dose optimization resulted in decreased mortality in several patients who participated in the trials [37].

### 2.3. Oxazolidinone

Oxazolidinone is a class of antibiotics that is observed to have activity against Gram-positive bacteria and was originally used in treatment for *Staphylococcus aureus* infection [38]. The first oxazolidinone drug was developed to combat several plant diseases. Oxazolidinone-based antibiotics for human use were synthesized, but their usage was terminated due to observed toxicity properties. However, in the late 1990s scientists were able to synthesize the non-toxic derivate of oxazolidinone, i.e., linezolid, and this drug is used to treat a wide-range of bacterial infections [39]. 

Oxazolidinone drugs mainly interfere with MTB through binding to the 30s and 50s subunit of the ribosome in the bacteria. This action will cause disruption in protein synthesis in MTB cells because the inhibition will result in a shorter amino acid chain and disturb the translation process, thus causing bactericidal activity [40]. A study of the structure of oxazolidinone drugs revealed that the N-aryl and 5-acylaminomethyl groups are the essential part in the oxazolidinone compounds to gain their activity. The electrophile group will increase the affinity toward the target, and the other, additional, aromatic ring in the oxazolidinone structure affects the pharmacokinetics of the drug [41].

Linezolid was characterized as a promising candidate to treat drug-resistant TB and is currently classified by the WHO as group A drug to treat MDR-TB and XDR-TB [42]. However, the efficacy of linezolid usage is limited due to its side effects such as anemia and neuropathy. This toxicity effect resulted in dose limitation in linezolid-containing regimens that are usually applied to long-duration MDR-TB treatment [43]. Additionally, MTB’s susceptibility to linezolid is decreasing due to acquired mutation in about 10% of the MDR-TB strains. Studies revealed that mutation in the *rrl* and *rplC* genes in MTB is responsible for linezolid resistance [44].

There are several oxazolidinone-derived compounds that are currently on trial for TB treatment regimens, such as delpazolid, sutezolid, and TBI-223. Delpazolid and sutezolid are currently in phase 2 and TBI-223 in phase 1 clinical trials. Delpazolid is a second-generation analog of oxazolidinone, containing a cyclic amidrazone group [45]. From the preclinical study, delpazolid has been characterized to have a role in binding to domain V of the 23s rRNA, thereby resulting in the inhibition of bacterial protein biosynthesis [46]. Sutezolid is a thiomorpholinyl analog of linezolid and has showed promising activity in MDR-TB treatment. Sutezolid is sought to shorten the regimen since it showed faster bactericidal activity than linezolid in the in vivo model [47]. Both delpazolid and sutezolid have passed the evaluation for safety and pharmacokinetics in the first phase. Another oxazolidinone in the trial is TBI-223. TBI-223 is a novel oxazolidinone that has shown high oral bioavailability, no CYP (Cytochrome) induction, and satisfying activity against MTB in a preclinical trial [48]. The structures of the oxazolidinones in TB drug trials are visualized in Figure 4. 

### 2.4. Nitroimidazole

Delamanid and pretomanid (Figure 5), two recently approved drug for MDR-TB treatment, belong to the nitroimidazole class. Since the discovery of metronidazole, the first class of nitroimidazole in the mid-1950s, nitroimidazole has already become known for its usage in antibacterial treatment and is widely employed as a treatment for infectious diseases [49]. Metronidazole is used to treat several bacterial diseases such as dysentery, peritoneal infections, colitis, and stomach ulcer [50,51]. Nitroimidazole enters cells through diffusion and is reduced to form several reactive nitrogen species, such as the nitro anion radical and nitroso and hydroxylamine derivatives. These compounds can cause DNA degradation and strand wreckage [52]. Nitroimidazole absorption is high in oral administration, with a relatively low binding rate to protein. Nitroimidazole is metabolized in the liver and excreted through urine [53].

Nitrogen-containing groups in a compound are usually related to the toxicity properties of a drug candidate. Since nitroimidazoles are heterocyclic nitrogen-containing nitrogen atoms, they require enzymatic reduction in order to summon their therapeutic effect, and this process can lead to the production of several mutagenic, carcinogenic, and hepatoxic species [54]. The efforts to minimize the toxicity properties of the nitroimidazoles have been carried out through several approaches such as bioisosteric replacement of the nitro group with another group such as benzoyl, acetyl, or amide, and using the nitro analog compounds while preserving the needed activity. Unfortunately, both approaches resulted in diminished in vivo activity of the drug candidate [55].

Although the existence of the nitro group in a compound is usually translated into the mutagenic profile, delamanid and pretomanid did not exhibit mutagenicity characteristics in preclinical and clinical trials [56,57]. From a study, it has been concluded that the position of each substituent surrounding the nitro group has a prominent role in determining the mutagenic characteristics [42] of a nitroimidazole compound. Furthermore, the study discovered a novel class of nitroimidazole, i.e., benzothiazole-based compounds, that are non-mutagenic [58].

Delamanid and pretomanid both target the cell wall synthesis pathway in MTB. Delamanid is a prodrug that needs to be activated enzymatically using deazaflavin-dependent nitroreductase [59]. Delamanid acts through the inhibition of methoxymycolic and ketomycolic acid (mycobacterial cell wall components) synthesis [60]. Similar to delamanid, pretomanid also targets cell wall synthesis by inhibiting the mycolic acid synthesis pathway. Additionally, pretomanid has another bactericidal action, which is respiratory poisoning of MTB. Pretomanid is able to release toxic nitrogen oxide within mycobacterial cells [61]. Nitric oxide (NO) can target multiple enzymes in the MTB cell including ATP synthase, Pks13, and RNA polymerase [62]. Both delamanid and pretomanid exhibited favorable activity through pharmacodynamics and pharmacokinetics profiling in the preclinical and clinical trials [63]. However, several cases of resistance to delamanid were reported due to mutations on the gene coding the delamanid nitroreductase activator [64].

### 2.5. Diarylquinoline

This class is categorized as a novel antitubercular chemical due to the discovery of bedaquiline—a first-line drug that has been recently approved to treat DR-TB [65]. Bedaquiline is the first newly approved drug for TB in more than 40 years, and this compound exhibited satisfactory results in a phase IIB clinical trial with placebo. Treatment duration was also reduced [66]. Bedaquiline acts as an antimycobacterial by targeting the ATP synthase pathway and terminating the energy generation pathway, leading to cell death [67]. Although bedaquiline showed a promising therapeutic effect, the safety of this drug remains a concern, and usage of this drug is limited to the treatment of pulmonary DR-TB. 

Bedaquiline was previously known as R207910 and TMC207 [68]. This compound has a planar hydrophobic part and groups of hydrogen bond acceptor/donor pairs. The hydroxyl group and N,N-dimethyl(-N(CH_3_)_2_) represent the hydrophobic part of this compound and are responsible for binding to the drug target, ATP synthase (Figure 6). The stability of diarylquinoline compounds is highly determined by the hydrogen bonding formed within the interaction [69]. A molecular biology study has confirmed that the main binding site for this drug is subunit c in the C ring of MTB ATP synthase, and the specificity for this drug is high because several mycobacteria have a mutation around Tyr64. This mutation caused binding site alteration [70]. However, after the approval of bedaquiline use, several cases of resistance were observed in the patients in treatment. Resistance to bedaquiline is caused by the mutation of genes such as *atpE*, *Rv0678*, and *pepQ*. The gene *atpE* codes for ATP synthase, the primary target of bedaquiline, while *Rv0678* acts to regulate the efflux pump expression, and the mode of action *pepQ* is still not fully known [71]. 

TBAJ-587 and TBAJ-876 are two novel diarylquinolines, the analogs of bedaquiline. Both compounds are now being tested in a preclinical trial to determine their suitability as drug candidates. TBAJ-576 is the 6-cyano analog of bedaquiline, and it showed promising efficacy toward MTB [72]. TBAJ-876 is the 3,5-dialkoxypyridine analog of bedaquiline and is characterized as an alternative to bedaquiline. TBAJ-876 exhibited submicromolar activity in vitro against mycobacteria but is less lipophilic than bedaquiline, resulting in a much safer compound [73]. The structures of bedaquiline, TBAJ-587, and TBAJ-876 are visualized in Figure 7. 

### 2.6. Benzothiazinone

Benzothiazinone is a sulfur-containing heterocycle compound. This chemical class was known by its antitubercular activity in vitro, in vivo, and ex vivo against MTB. A structure–activity relationship study revealed that the sulfur atom and nitro group contained in these compounds were important for its bactericidal activity [74].

The DprE1 (decaprenylphosporyl-β-D-ribofuranose-2′-epimerase) enzyme is an essential enzyme in MTB’s life cycle, as it is responsible for the biosynthesis of arabinans. Arabinogalactan is one of the vital components of the mycobacterial cell wall [75]. DprE1 and DprE2 work simultaneously to conduct the epimerization of decaprenylphosporyl- β-D-ribofuranose to decaprenylphosporyl-β-D-arabinose, the precursor for arabinan. The nitro group in benzothiazinone is converted to a reactive nitroso-group by FADH_2_ in DprE1 that further binds specifically to C387 in the DprE1 active site irreversibly [76,77]. Benzothiazinone acts as an inhibitor for DprE1, interfering with the MTB cell wall formation, which makes MTB susceptible to various kinds of bactericidal agents [78]. 

BTZ-043 and macozinone (PTBZ-169) are two benzothiazinones that are now in a clinical trial for DR-TB drugs (Figure 8). BTZ-043 was the first benzothiazinone characterized for its activity toward MTB, with a minimum inhibitory concentration (MIC) of 1 ng/mL, significantly more effective compared to isoniazid or rifampicin [79]. The lead optimization process for BTZ-043 resulted in the discovery of macozinone, with an MIC of 0.3 nM, making this compound the most potent benzothiazinone against MTB [80]. Both BTZ-043 and macozinone are active toward MDR- and XDR-TB strains and are considered safe [81]. Several backup studies to develop another antitubercular benzothiazinone compound with better bioavailability and stability were conducted to improve the pharmacodynamics and to anticipate probable mutations in the vital spot of the DprE1 active site [80,82].

### 2.7. Other Classes

In this section, drug candidates for TB in clinical trials that do not belong in any of abovementioned classes will be reviewed.

Telacebec, which was previously known as Q203, is a derivative compound of imidazopyridine amide that is currently in the second phase of a clinical trial. This compound showed activity toward MTB reference strain H37Rv better than other first-line approved drugs, making telacebec a compound of interest in TB drug discovery [83]. Telacebec exhibits acceptable adverse-event rates and satisfactory results in the clinical trial [84]. Similar to diarylquinoline compounds, imidazopyridine amide compounds target the mycobacterial energy metabolism as their mechanism of action. MTB growth is greatly affected by the respiratory chain and ATP synthesis as its energy production processes [85]. The cellular target of telacebec is the cytochrome bcc complex, which belongs to complex III in the mycobacteria respiratory chain. This protein is coded by the *qcrB* gene in MTB. Resistance toward this chemical class was observed in a strain carrying a T313I mutation on *qcrB* [86].

### 2.8. Drug Repurposing for TB Treatment

Drug repurposing is one approach that is used to discover a novel antituberculosis agent. This effort is not only time and cost effective but it also minimizes the possibility of cross-resistance as the targets of these known drugs are probably novel targets in MTB [87]. Nitazoxanide and sanfetrinem are two antituberculosis drug candidates that were discovered through a drug repurposing approach (Figure 9).

Nitazoxanide (2-acetyloxy-N-(5-nitro-2-thiazoyl)-benzamide) is a nitrothiazolyl-salicylamide derivate that firstly known as a broad-range antiprotozoal agent. This compound works as a pyruvate ferredoxin oxidoreductase enzyme inhibitor, and later it was confirmed that nitazoxanide exhibited activity against several anaerobic bacteria with the same cellular target [88]. Through a drug repurposing study, nitazoxanide’s bactericidal activity toward the MTB reference strain was confirmed [89]. In MTB, nitazoxanide works through membrane potential and pH homeostatic disruption [90]. Nitazoxanide is currently in phase 2 of a clinical trial.

The second antituberculosis drug candidate from the drug repurposing effort in the preclinical trial is sanfetrinem. Sanfetrinem is a beta-lactam compound, and its antitubercular activity was found through the screening of thousands of beta-lactams against MTB. Sanfetrinem exhibited the most active bactericidal result against H37Rv MTB among all compounds and is known for its record of clinical safety [91]. This beta-lactam compound interferes with peptidoglycan biosynthesis in MTB, inhibiting the cell wall formation process [92].

Another compound in a phase 2 clinical trial for TB drugs is SQ109. SQ109 belongs to the same class of compounds as the previously known TB drug ethambutol, i.e., ethylenediamine [93]. However, unlike ethambutol, SQ109 has a different mechanism of action. SQ109 targets MmpL3, the membrane transport protein (Rv0206c) in MTB. This protein is highly conserved within the mycobacteria. This protein acts as a trehalose monomycolate transporter, one of the MTB cell wall components [94].

TBA-7371 and OPC-167832 (Figure 10) are two novel DprE1 inhibitor compounds. Both TBA-7371 and OPC-167832 do not belong to the same class as another DprE1 inhibitor (macozinone and BTZ-043). TBA-7371 is 1,4-azaindole compound that can bind noncovalently to DprE1. Azaindoles were first identified as a promising chemical class for TB drug candidates after they showed in vivo MTB bactericidal activity through a series of scaffold-morphing studies [95]. TBA-7371 did not exhibit any cross-resistance with other covalently bound DprE1 inhibitors and was characterized to be similarly active against both drug-susceptible or drug-resistant MTB strains [96]. OPC-167832 is a derivate of 3,4-dihydrocarbostyril that showed potent antitubercular activity starting from a 0.625 mg/kg dose [97]. Both TBI-7371 and OPC-167832 are currently in phase 2 clinical trials. Other compounds with different chemical classes such as SPR-720 (ethyl urea benzimidazole), TBI-166 (riminophenazine), and Spectinamide-1810 (spectinamide) are drug candidates in the early phase of trials for TB treatment [98,99,100]. All the drugs discussed above are summarized in Table 1. 

### 2.9. Host-Directed Therapy

Upon pathogen infection, the innate and adaptive immune system of the host plays important roles to decide the interaction. Dependent on several host-related factors, the host–pathogen interaction can result in containment of the infection or disease progression, which could lead to either recovery or fatality [119]. Until now, with all the discovered chemotherapy agents, human TB is still a global threat with high fatality. Current TB therapy has several limitations such as long duration, complicated regimen, and drug toxicity. Moreover, the occurrence of DR-TB has brought up the urgency to develop another approach to contain this disease [120]. Host-directed therapy (HDT) is a novel approach that can act as an additional strategy for TB treatment. HDT principles are to interfere with any essential host–pathogen interactions related to pathogen replication and to enhance the host immune response through stimulation of host factors or another immunogenic component. HDT can be achieved using small molecules/biologics [121]. The usage of HDT for DR-TB treatment can open the possibility of a shorter-time, more effective treatment without worrying about the possibility of therapy resistance because the compound will target the conserved host [122].

The importance of immune response during MTB infection was clarified by the development of latent TB, where an immunocompetent patient will show no symptoms. Latent TB is characterized through a dynamic replication equilibrium of MTB in the host tissues. This mechanism occurs when the host can suppress the replication rate of MTB, and this condition could last for a long time [123]. Host response to TB infection is a complex process involving the innate and adaptive immune system. Although each component in the immune system has a prominent role, macrophages, CD4+ T lymphocytes, and granuloma formation are considered the main player in the defense mechanism [124]. With an adequate knowledge of the host response during the infection, several target pathways for TB HDT have been characterized, such as autophagy, cholesterol, eicosanoids, and granuloma. The mechanism for each target could differ according to the HDT compounds [125]. Several studies have resulted in the discovery of HDT compounds, whether it is a novel compound or from a repurposing effort. Three compounds that are expected to undergo trial for DR-TB HDT are CC-11050, everolimus, and cholecalciferol. Both everolimus and cholecalciferol are already well known, and CC-11050 is a novel anti-inflammatory compound. 

CC-11050 acts as a phosphodiesterase-4 (PDE4) inhibitor. PDE4 plays a role in cyclic nucleotide hydrolysis, specifically of cyclic adenosine monophosphate (cAMP) present in the host leukocytes such as macrophages, neutrophils, and lymphocytes [126,127]. By blocking the hydrolysis process, the inflammation also can be regulated negatively [128]. CC-11050 showed potential in a preliminary TB study with a rabbit model and coadministration with INH and is currently on trial preparation for DR-TB HDT [129]. In addition to the antituberculosis properties, this compound has gone through clinical trials for other diseases such as HIV and lupus [129,130].

The second HDT candidate for DR-TB therapy is everolimus. Everolimus is an inhibitor for the mTOR (mechanistic target of rapamycin) serine/threonine kinase signal transduction pathway [131,132]. This pathway regulates growth through the control of most metabolic process related to nutrients and nutrient-induced signals. The mTOR inhibitors are commonly used as immunosuppressant and anticancer agents [133,134]. Although the mechanism is still unclear, autophagy is the main proposed target for everolimus TB HDT. mTOR inhibition has been associated in autophagy inducement in the granuloma, protecting the host from MTB infection [135]. The usage and safety of everolimus for TB HDT still needs to be tested in clinical trials, following previous reports of reactivation of latent TB in a metastatic renal cell carcinoma patient who was treated with this compound [136].

Cholecalciferol is commonly known as vitamin D and used as a dietary supplement. In the human body, this compound is recognized as a regulator of calcium homeostasis and believed to have an anti-inflammatory effect [137]. Vitamin D acts as the promoter of macrophages and monocytes in the human immune system, resulting in the restriction of MTB growth [138]. Conversion of vitamin D to its active form, 1,25-dihydroxycalciferol, involves the activation of CYP27B1 by TLR2 or INF-ɣ, which generates antimicrobial peptides. In addition to this proposed mechanism, vitamin D has been characterized to induce autophagy through LL-37 production [139]. As several studies reported that vitamin D deficiency is observed in TB patients, the usage of vitamin D to treat TB as an adjunctive therapy has been on trial ever since [140,141].

Tuberculosis and HIV (human immunodeficiency virus) comorbidity has the potential to accelerate the deterioration of immunological functions, resulting in premature death if untreated. The increased incidence of active TB in HIV-infected individuals can be attributed to at least two mechanisms: the reactivation of latent TB or increased susceptibility to MTB infection [131,142]. 

### 2.10. Tuberculosis (TB) Immunotherapy

Immunotherapies could regulate the immune system in patients with latent TB infection or active disease, enabling better control of MTB replication. Immunotherapies ideally should modulate the immune system in a role that helps the host control or eliminate MTB. There are several types of essential immunotherapies with a focus on those that have been tested in human on clinical trials (Table 2) [143].

The RUTI vaccine is among the few candidates that are currently in the clinical trials to be specifically developed as a therapeutic TB vaccine. The vaccine is composed of purified and liposomal cellular fragments of MTB bacilli cultures under stress to mimic intra-granulomatous conditions and to induce latency antigens, which would typically be hidden from the immune system. The immune response to RUTI is characterized by a poly-antigenic response. Its main immunotherapeutic effect is to induce T helper-1 (Th1) response, not only against growth-related antigens but also against structural antigens. The prophylactic proportion of the RUTI vaccine has also been evaluated in a murine model and could significantly reduce bacterial counts in both the lung and the spleen [156]. 

## 3. Challenges

Successful usage of the first-line drugs for drug-susceptible TB treatment has given rise to another problem, i.e., the development of drug-resistant TB (DR-TB). This occurrence can be observed as the consequence of MTB’s natural evolutionary process and standardized treatment regimens regardless of the emergence location, severity, pathology, or comorbidities [157]. This occurrence resulted in treatment failure since the treatment for DR-TB is more complicated than that for drug-susceptible TB. The WHO Global Tuberculosis Report 2018 listed that the success rate of DR-TB treatment in 30 high TB burden countries was around 48–86% with an average of 56%, with the lowest success rate percentage observed in India, Indonesia, Mozambique, and Ukraine [1]. This finding indicated that there is still room for improvement in the standardized protocol for DR-TB treatment. This section will discuss more about the challenges faced in DR-TB treatment. 

Rapid and accurate diagnosis is one of the challenges in DR-TB treatment. A recent projection study for DR-TB emergence resulted in the possibility of a massive increase in MDR-TB cases between 2015 and 2025. This finding was supported by another fact that if by 2017 all patients with TB could be tested for resistance, the MDR-TB incidence in 2025 will be reduced by 29% [158]. There are several notable diagnostic tests for DR-TB from the traditional culture technique to the more sophisticated molecular techniques, Xpert MTB/RIF, GeneXpert, and whole-genome sequencing, but not all people have access to these facilities due to financial or location factors. Another limitation for these diagnostics tools is the ability to detect mutation outside the commonly known region (for example, the rifampicin-resistance determining region in the *rpoB* gene), which happens in a minor proportion of DR-TB cases, not to mention the time-consuming diagnosis and costly equipment [159]. Every year, it is estimated that there are more than three million missed TB cases [160].

The long duration and complicated drug regimen also have contributed to the challenges facing DR-TB treatment, as they influence treatment adherence. Treatments for MDR-TB and XDR-TB often demand 6–18 months of unstopped therapy, with multidrug administration. The treatment has to consider the previous medications taken by the patient, along with the dose and duration, to determine the suitable drug regimen [13]. In addition to the complexity, DR-TB treatment is also inhibited by the cost factor, where the therapy often costs up to 25 times higher than the cost for drug-susceptible TB treatment [161]. These factors are often caused in untreated cases due to various stigma in society, socioeconomical burden (DR-TB patients usually lose their income due to the long duration of the therapy), and lack of knowledge [162]. Treatment adherence plays a prominent role in TB therapy, as incomplete therapy can increase failure risk and result in disease relapse, further transmission, development of even-worse drug resistance, or even fatality [163]. From a recent trial, the adherence threshold was defined as 76–80% of intended doses taken by the patient [164]. Several factors such as forgetting to take the medication, missing the appointment date, inability to cover the cost, and lack of medicine often cause DR-TB patients to not obey the long-course DR-TB treatment. As an example, India reported non-adherence to treatment for more than 50% of the patients [165].

In addition to the diagnostics and long-duration treatment problem, side effects exhibited by the drugs contribute to the inefficiency of the treatment. The drug usage in the treatment regimen should be reviewed and discontinued every time uncontrolled side effects are observed in patients. The side effects are mainly ototoxicity, nephrotoxicity, hypothyroidism, psychiatric disorders, epileptic episode, and uncontrolled gastrointestinal disturbance [166]. Usage of bedaquiline and delamanid, two recently approved drugs for DR-TB treatment, requires electrocardiograph monitoring to prevent the development of unwanted side effects in high-risk patients [14].

The drug discovery effort to find a highly effective drug with low side effects for DR-TB treatment is hindered by several factors, mostly related to the physiological nature of MTB. The lead compounds from high-throughput screening usually failed in the next step when they have to be tested against the pathogen using a whole-cell MTB assay, leading to a high failure rate for the drug discovery and development for this disease [167].

## 4. Conclusions

As number of DR-TB cases emerge each year, there are several antitubercular agents that have been discovered. These natural and synthetic compounds belong to various chemical classes and are each tested in trials to ensure their safety and bactericidal activity. However, considering several challenges in DR-TB treatment (diagnostics failure, long-duration treatment that influences treatment adherence, and limitations in drug discovery due to MTB’s nature), the host-directed therapy approach in the DR-TB treatment regimen should be considered.

## Figures and Tables

**Figure 1 pharmaceutics-13-00592-f001:**
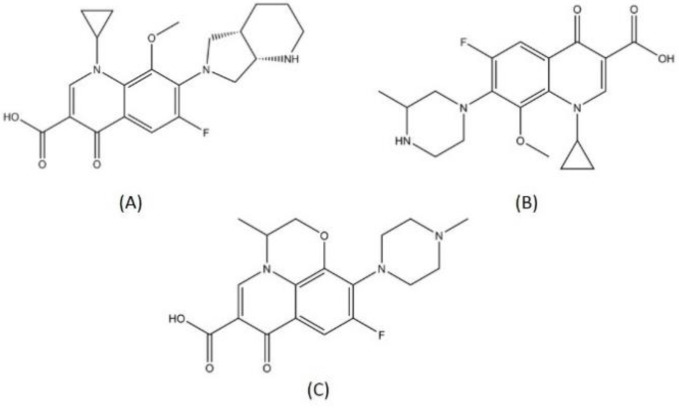
Structure of (**A**) moxifloxacin, (**B**) gatifloxacin, and (**C**) levofloxacin.

**Figure 2 pharmaceutics-13-00592-f002:**
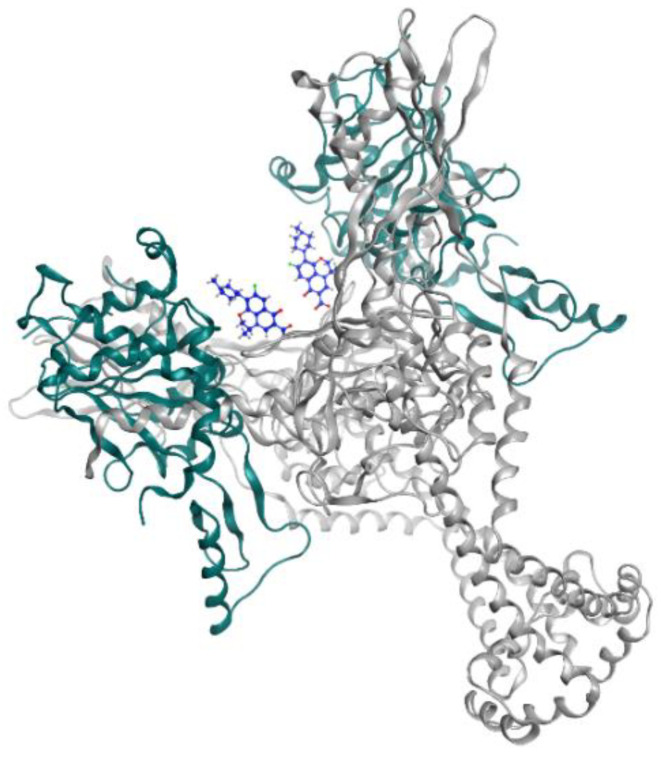
Levofloxacin (blue) in DNA gyrase subunit A (gray ribbon) and subunit B (green ribbon) (PDBID: 5BTI).

**Figure 3 pharmaceutics-13-00592-f003:**
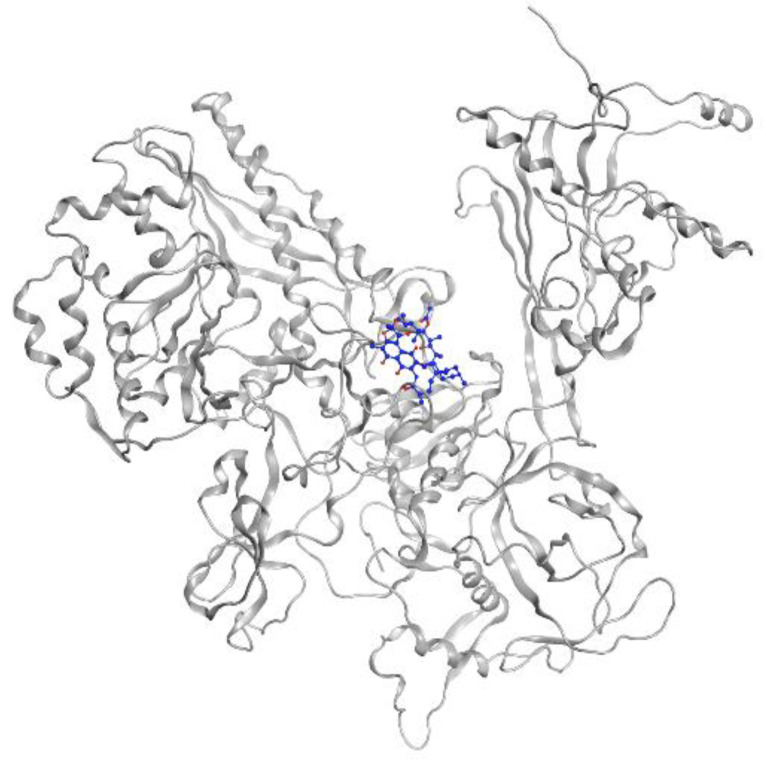
Rifampicin (blue) in MTB RNA polymerase subunit β (gray ribbon) (PDBID: 5UH6).

**Figure 4 pharmaceutics-13-00592-f004:**
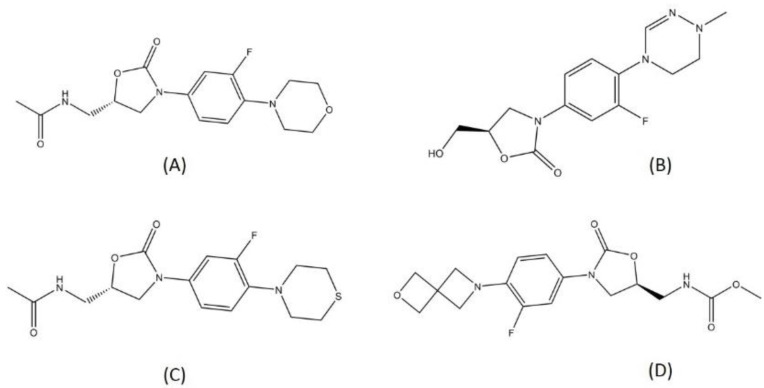
Structure of (**A**) linezolid, (**B**) delpazolid, (**C**) sutezolid, and (**D**) TBI-223.

**Figure 5 pharmaceutics-13-00592-f005:**
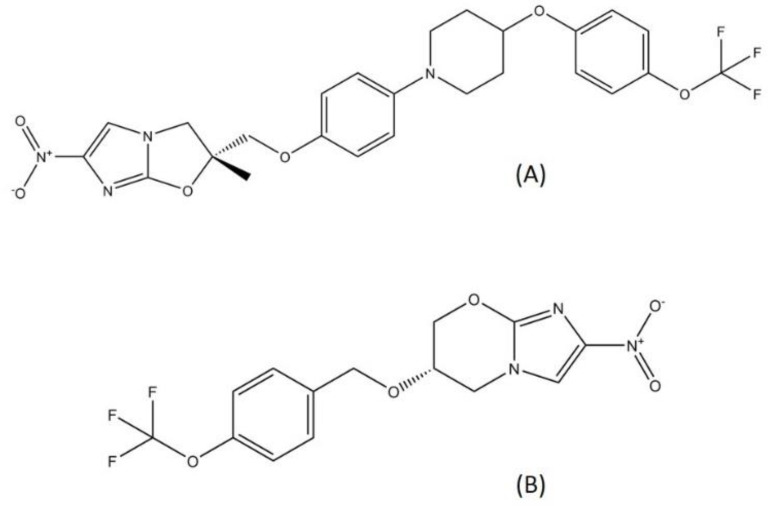
Structure of (**A**) delamanid and (**B**) pretomanid.

**Figure 6 pharmaceutics-13-00592-f006:**
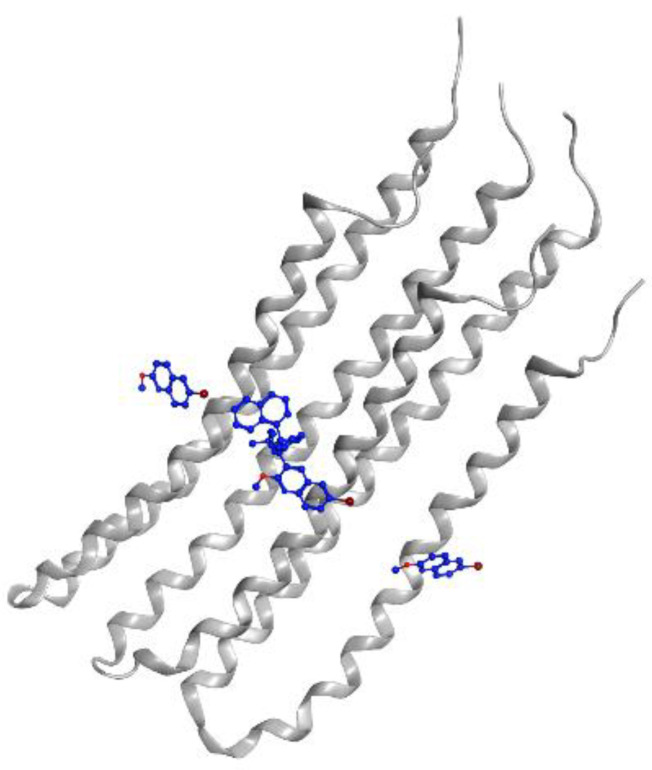
Bedaquiline (blue) in mycobacterial ATP synthase subunit C (gray ribbon) (PDBID: 4V1F).

**Figure 7 pharmaceutics-13-00592-f007:**
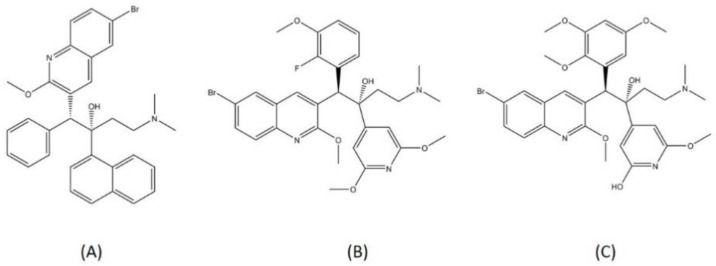
Structure of (**A**) bedaquiline, (**B**) TBAJ-587, and (**C**) TBAJ-876.

**Figure 8 pharmaceutics-13-00592-f008:**
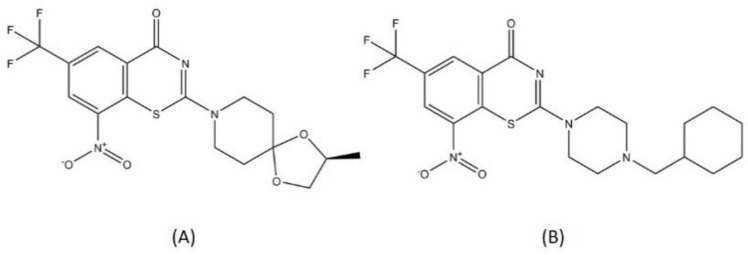
Structure of (**A**) BTZ-043 and (**B**) macozinone.

**Figure 9 pharmaceutics-13-00592-f009:**
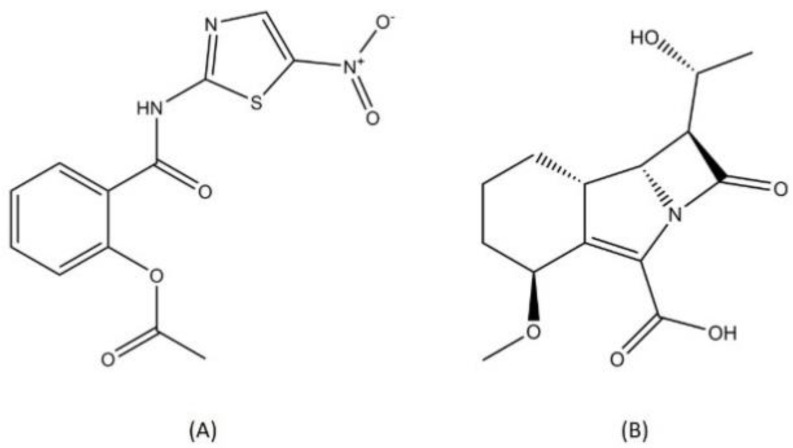
Structure of (**A**) nitazoxanide and (**B**) sanfetrinem.

**Figure 10 pharmaceutics-13-00592-f010:**
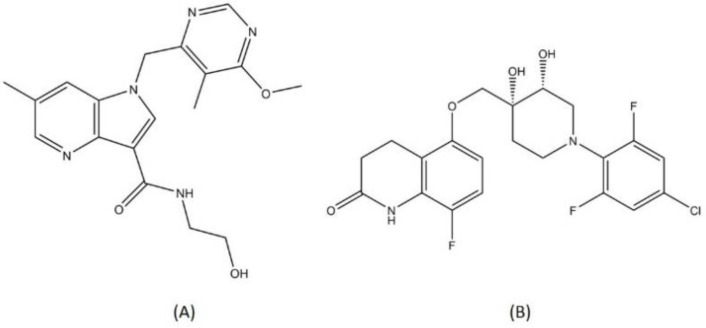
Structure of (**A**) TBI-7371 and (**B**) OPC-167832.

**Table 1 pharmaceutics-13-00592-t001:** TB drugs in clinical trials [101].

Chemical Class	Compound	Progress	Mode of Action	Reference(s)
Fluoroquinolone	Levofloxacin	Phase 2	DNA gyrase inhibitor	[102,103]
Rifamycin	Rifampicin (high dose)	Phase 2	RpoB inhibitor	[37]
Oxazolidinone	Delpazolid	Phase 2	Inhibition of protein synthesis	[104,105]
	Sutezolid	Phase 2	Inhibition of protein synthesis	[47]
	TBI-223	Phase 1	Inhibits the binding of N-formylmethionyl tRNA to ribosome	[106]
Nitroimidazole	Delamanid	Phase 3—approved	Inhibits cell wall synthesis	[107]
	Pretomanid	Phase 3—approved	Inhibits cell wall synthesis	[108]
Diarylquinoline	Bedaquiline	Phase 3—accepted	Inhibits mycobacterial ATP synthase	[109]
	TBAJ-587	Preclinical trial	Inhibits mycobacterial ATP synthase and hERG potassium channel	[110]
	TBAJ-876	Preclinical trial	Inhibits mycobacterial ATP synthase	[73,111]
Benzothiazinone	Macozinone	Phase 2	DprE1 inhibitor	[112,113]
	BTZ-043	Phase 1	DprE1 inhibitor	[74]
Other classes	Telacebec (imidazopyridine)	Phase 2	QcrB inhibitor	[84]
	Nitazoxanide (nitrothiazolyl-salicylamide derivate)	Phase 2	Disruption of membrane potential and pH homeostasis	[114,115]
	SQ109 (ethylenediamine)	Phase 2	MmpL3 inhibitor	[37,94]
	TBA-7371 (1,4-azaindole)	Phase 2	DprE1 inhibitor	[116]
	OPC-167832 (3,4-dihydrocarbostyril derivate)	Phase 2	DprE1 inhibitor	[117]
	SPR-720 (ethyl urea benzimidazole)	Phase 1	GyrB inhibitor	[98]
	TBI-166 (riminophenazine)	Phase 1	Membrane destabilization	[99]
	Sanfetrinem (beta-lactam)	Preclinical trial	Inhibits peptidoglycan synthesis	[118]
	Spectinamide-1810 (spectinamide)	Preclinical trial	Selective ribosomal inhibition	[100]

**Table 2 pharmaceutics-13-00592-t002:** Immunomodulating immunotherapies for treatment of TB in humans.

Therapeutics	Composition	Target (Outcome)	References
*Mycobacterium vaccae*	Killed, interdermal	Meta-analysis of 54-studies on newly diagnosed pulmonary TB (improved sputum conversion and X-ray changes	[143,144]
	Capsule	Faster smear conversion	[143,145]
RUTI^®^	Detoxified cellular fragments of *Mycobacterium tuberculosis*	Phase I and II clinical trials on LTBI cases or healthy volunteers (immunogenic, reasonable tolerability)	[143,146,147]
Autologous MSC	MSC	MDR or XDR patients (with radiologic improvement)	[143,148]
V5 immunitor	Inactivated pooled blood	Re-treatment or proven MDR (higher rate of sputum conversion)	[143,149]
Cytokines and cytokine inhibitors	IL-2	MDR-TB patients (better sputum conversion rate), MDR-TB patients (decrease AFB smear counts with daily IL-2 compared to control or pulse IL-2), new TB patients (significant delays in culture conversion)	[143,150]
	IFN-γ	MDR-TB patients (all smear negative/improved, MDR-TB cases (no marked microbiologic effect), HIV-positive TB cases (more rapid culture conversion compared to historical control)	[143,151]
Drugs/compounds	High-dose steroid	HIV-positive TB cases (increased culture conversion at 1 month)	[143,152]
	Levamisole	Newly diagnosed pulmonary TB patients (improved radiology, but no effect on smear conversion)	[143,153]
	Albendazole	New pulmonary TB patients (no effect on clinical, radiologic, and microbiologic outcome)	[143,154]
	Thalidomide	HIV-positive (clinical improvement), HIV-positive (no clinical difference)	[143,155]

## Data Availability

Not applicable.

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
