# Peer review of "Recent Progress and Challenges for Drug-Resistant Tuberculosis Treatment"

_pharmaceutics, 2021, doi:10.3390/pharmaceutics13050592_

Round 1

Reviewer 1 Report

This paper presents a very interesting comprehensive overview of the current trends in development of recently introduced and upcoming antibiotics against drug resistant M. tuberculosis. I support publishing this paper. I have only several minor comments.

Lines 60-67. The discussion about the mechanisms of drug resistance development in Mtb is given in such a way that it can be concluded by readers that the horizontal gene transfer may play a role in the acquired resistance in Mtb. Reword this paragraph to avoid such misinterpretation. In this cited paper by Nguyen (2016) it was well highlighted: “Acquired antibiotic resistance may occur in bacteria through either mutations or horizontal gene transfer mediated by phages, plasmids or transposon elements. In M. tuberculosis, horizontal transfer of drug resistance genes has not been reported…”

Line 120: “Levofloxacin observed to has… “, should be “Levofloxacin was observed to have…”

Line 148: double “and”

Line 154: “Mutation occurs in the codon 51m 526,…”, should be “51, 526”. Is it really codon 51? This codon is outside the rifampicin-resistance determining region.

Line 170: gram-positive, should be Gram-positive.

Line 204: “no CYP induction”, CIP abbreviation should be deciphered.

Line 220: “Nitrogen group in a compound is usually related…”, should be “Nitrogen containing groups in a compound are usually related…”

Line 221: “Since nitroimidazoles are heterocyclic nitrogen…”, may be “heterocyclic compounds containing nitrogen atoms”

Line 243: “Nitrogen oxide” – specify which nitrogen oxide, there are several possible variants.

Line 288: “This chemical class was known its antitubercular activity…”, should be “was known by its antitubercular activity…”

Line 298: “Benzothiazinone acts as inhibitor for DprE1, interfering the MTB cell wall formation that makes MTB insensitive to various kind of bactericidal agent” – may be “susceptible to various kind of bactericidal agent”?

Line 488: what is cellThese? I Googled the term by found only some batteries.

Bacterial species name must be in italic: lines 9, 24, 106,

Gene name must be in italic: lines 96, 97, 105, 270, 272

Locus tag should not be in italic typeface: 270

Author Response

The authors have been revised according to the suggestions and comments of Reviewers. The detail has been attached in the file below

Reviewer 2 Report

“Recent Progress and Challenges for Drug-resistance Tuberculosis Treatment” – the content of the manuscript is not extensive as suggested by the title. The progress in the clinical trials is described, however only for individual compounds. The new treatment regimens aiming to combat the drug resistance are neglected/omitted. The new treatment regimens which are under clinical trials should be commented, because it is highly important that monotherapy is practically not applied for TB treatment.

What is more, the drug resistance in TB is much more complicated problem, not only related to drug resistance itself but also to immunity deficiencies in patients and great problem of HIV co-infection. The HIV co-infected tuberculosis is not commented in this manuscript, although is a big contribution to the TB treatment failure.

Authors developed the chapter about the Host Directed Therapy – this is very important strategy, however the topic is presented only to limited extent. This section should be much more elaborated because currently it is the most promising strategy. Also the Tuberculosis (TB) Immunotherapy being currently under Phase 2 Study should be presented. Additionally, the clinical trials were also conducted to check the plant derived compounds as boosters of immunity during TB treatment. This aspect is not less important, also because such compounds alleviate the adverse effects of classical therapy (eg. DOI: 10.2991/efood.k.200418.001; DOI: 10.5772/intechopen.83030). Also cancer complicated TB should be commented. Even WHO discusses immunotherapy (https://www.who.int/tdr/publications/documents/interventions-tb.pdf).

Other comments:

Drug repurposing should be described as separate section

Section2.3: The text relates only to the action of rifampicin, any advantages of its derivatives are nor presented, as the progress in clinical trials is not discussed for other compouds (only for higher doses of rifampicin)

Useful databases:

ClinicalTrials.gov

Cohrane database

Author Response

The authors have been edited and revised the manuscript according to the Reviewer's suggestions.

Round 2

Reviewer 2 Report

The remarks were addressed.